# Strain Monitoring and Installation Adjustment of Satellite–Rocket Connection Device Based on Distributed Optical Fibers

Xiaoxi Qu [1], Shiyuan Zhao [2], Fuqiang Ma [3], Jianle Li [1], Hanke Li [1], Zhengyan Yang [4,*], Hao Xu [5], Lei Yang [1,*] and Zhanjun Wu [5]

1   State Key Laboratory of Structural Analysis, Optimization and CAE Software for Industrial Equipment, School of Mechanics and Aerospace Engineering, Dalian University of Technology, Dalian 116024, China; syquxiaoxi@mail.dlut.edu.cn (X.Q.); lijianle@mail.dlut.edu.cn (J.L.); hank@mail.dlut.edu.cn (H.L.)
2   School of Optoelectronic Engineering and Instrumentation Science, Dalian University of Technology, Dalian 116024, China; zhaoshiyuan@dlut.edu.cn
3   Dalian Scientific Test & Control Technology Institute, Dalian 116013, China; ma_fu_qiang@126.com
4   College of Transportation Engineering, Dalian Maritime University, Dalian 116026, China
5   School of Materials Science and Engineering, Dalian University of Technology, Dalian 116024, China; xuhao@dlut.edu.cn (H.X.); wuzhj@dlut.edu.cn (Z.W.)
*   Correspondence: zyyang1993@dlmu.edu.cn (Z.Y.); yangl@dlut.edu.cn (L.Y.)

**Abstract:** In this study, a distributed optical fiber sensor was used for strain monitoring and installation adjustment of a satellite–rocket connection device under a preload. The distributed optical fiber sensor was installed on both the tape and the satellite docking frame, utilizing OFDR (Optical Frequency Domain Reflectometry) based on Backward Rayleigh scattering strain demodulation methods to precisely measure the strain distribution of both components when subjected to a preload. In order to deal with the uneven stress of the belt in the process of preloading, a finite element analysis was performed to obtain the strain distribution of the belt under preloading. The strain monitoring results of the optical fiber and strain gauge were compared, and the strain trend of the finite element simulation results was verified. Finally, the measured strain data were adopted to assist the installation and adjustment of the satellite–rocket connection device to achieve a uniform distribution of the preload. The experimental results showed that the standard deviation of strain at each position of the tape was reduced after adjustment. This study provides guidance for the installation of satellite–rocket connection devices.

**Keywords:** satellite–rocket connection device; strain monitoring; distributed fiber sensor; installation adjustment; preload

## 1. Introduction

The satellite–rocket connection device stands as a critical component for joining and separating launch vehicles and satellites. Its main function is to ensure the reliable connection of satellite and rocket during rocket-powered flight; after reaching the predetermined orbit, it ensures the reliable separation of the satellite and rocket according to the design requirements. The performance of the satellite–rocket connection device directly influences the success or failure of a rocket launch. The pivotal component within the satellite–rocket connection device is the clamp band, which, under a specific preload force, generates positive axial pressure through wedge blocks to ensure the dependable connection between the satellite and the carrier rocket. This process is achieved by loading a pre-tensioning force on the clamp band [1].

When the satellite and rocket separate, the explosion bolt securing the clamp band breaks, leading to the release of the clamp band's preload force. Simultaneously, aided by components like springs, the band retracts onto the carrier docking segment, thus

effectuating the separation of the rocket and satellite. V-bands offer advantages over traditional fasteners because they are easier to assembly and disassemble, particularly in confined spaces [2]. They are now widely utilized in the turbocharger and exhaust system, and in the marine, aerospace, pump, and pharmaceutical industries.

At present, researchers from various countries have matured their research on the theory and performance of bands, and there are many key technical issues are involved in the design and dynamic research of band connection structures. Robert et al. pointed out the inadequacy of using the radial slip between the V-block and the end frame as a criterion for clamp band connection failure. They proposed a new criterion for clamp band connection failure based on the gap between the satellite–rocket docking frames [3]. Jacob et al. established both a 3D and axisymmetric finite element model of the clamp band, and based on this, investigated the effects of frictional forces and preload forces on the bearing capacity of the clamp band [4]. Ladisa et al. used a nonlinear finite element model to investigate the contact issues of the clamp band connection device under working conditions. The results indicated a relatively severe jamming problem near the joint under the pre-tensioned state, leading to stress concentration at the joint [5]. Qin et al. analyzed the deformation characteristics of a certain type of wrapped connection device within a certain axial tension range based on elastic mechanics theory, and derived an analytical expression for stiffness. Guo et al. analyzed the nonlinear static force of the clamp band [6]. At the same time, the finite element method was used to analyze and calculate the axial bearing capacity, stiffness, and damping characteristics of the wrapped connection device. The experimental results verified the correctness of the theoretical analysis and finite element simulation [7–9]. Barrans et al. further considered the effects of friction and torque, and found that the distribution of contact force on the contact surface of the band connection device was uneven [10–12].

Due to the difficulty in conducting theoretical analyses of clamp band performance, the relationship between the friction force, preload force, and clamp band tension is very complex, making it difficult to establish mathematical relationships. In order to understand the force situation and laws of clamp band tightening devices, researchers primarily rely upon experimental analyses and validation. NASA also recommends using strain gauges to monitor the circumferential load distribution [1]. Ali et al. proposed an empirical method to determine the axial load distribution on the clamp band. This method measures the load using strain gauges and pressure sensors, and it aligns with the results of theoretical studies and finite element research on uneven load distributions [2]. Zhang et al. found through experiments that the loading position and gap arrangement can also affect the performance of the wedge effect, leading to deviations in contact pressure from the ideal distribution [13].

At present, in the final assembly process of some spacecraft, the preloading operation of the separation clamp band for the satellite launcher is carried out using mechanical loading devices. The separation clamp band is pulled by rotating the nut using a mechanical wrench, manually loading the pre-tightening force. This entirely relies on the cooperation of operators to achieve synchronous loading at various points and it must be locally adjusted to ensure even loading. However, this method results in low labor efficiency and unstable quality. Therefore, selecting and arranging suitable sensors to conduct real-time monitoring of stress and strain on the clamp band ping and satellite frame during the preloading process is of utmost importance. Fiber optic sensors have many advantages, including resistance to electromagnetic interference, multiplexing capabilities, a small size, and light weight. Therefore, in recent years, fiber optic sensing technology has been widely applied in the aerospace field [14]. NASA conducted experiments using FBG (Fiber Bragg Grating) for deformation measurements on the wings of unmanned aerial vehicles [15]. OFDR technology, based on Backward Rayleigh scattering, boasts high sensitivity, exceptional spatial resolution, multipoint measurement capabilities, and versatile applicability. These attributes position it with promising prospects for the widespread utilization in the field of fiber optic sensing for strain measurements. Distributed fiber optic sensing

with wavelength division multiplexing can measure strain with millimeter-level spatial resolution, and a single fiber can be used for multiple strain measurements [16,17]. INTA (Instituto Nacional de Técnica Aeroespacial) carried a fiber optic monitoring system on the OPTOS (Optical Nano-Satellite) to monitor the strain and temperature of the satellite in orbit [18]. The United States has developed the Peripheral Docking Mechanism LIDS (Low Impact Docking System) and established the International Low Impact Docking System standard, also known as the NASA Docking System (NDS) [19–21]. However, there is relatively little research on the stiffness and mechanical behavior of satellite docking frames, and their designs often use empirical values. Furthermore, these structures are elongated, narrow, and small, being enclosed beneath the clamp band, making it difficult to conduct monitoring. This to some extent affects the research on lightweight, high-load locking structures.

In conclusion, to address issues with traditional strain gauge monitoring methods such as fewer measurement points, potential strain information loss, and the inability to synchronously monitor the satellite frame, this paper proposes the deployment of distributed fiber optic sensors on the clamp band and satellite docking frame. During the installation of the connecting device, it monitors the strain on the clamp band and satellite docking frame under the action of preloading. A comparison was made between the strain monitoring results obtained from the optical fibers on the clamp band and strain gauges. A method is proposed that utilizes strain data on the clamp band to assist in installation debugging, achieving a uniform application of preloading force on the clamp band.

## 2. Methods

### 2.1. Description of the Clamp Band System

The clamp band-type satellite–rocket connection device (referred to as a satellite–rocket connection device) is a common connection method used between carrier rockets and payloads. It accomplishes the locking and unlocking functions of the carrier rocket and payloads such as satellites through the clamp band-type satellite–rocket connection device. The satellite–rocket connection device mainly consists of clamp band, blocks, tension springs, and explosive bolts, as shown in Figure 1. The docking frame comprises satellite and rocket frames and the clamp band is generally made of high-strength steel or titanium alloy, while the V-shaped blocks are made of aluminum alloy [22]. During locking, the satellite–rocket connection device provides a pre-tensioning force through explosive bolts. The blocks clamp the docking frame under the action of the pre-tensioning force, ensuring a reliable connection between the payload and the carrier rocket. When unlocking, the explosive bolts separate, releasing the pre-tensioning force. The docking frame, serving as the main energy storage structure, releases the stored deformation energy, providing initial kinetic energy for the separation of the interstage locking device. During the rocket's powered flight phase, lateral and longitudinal acceleration environments subject the interstage connection interface to tensile forces, moments, shear forces, and other loads. These loads may lead to structural damage to load-bearing components, such as plastic deformation of the docking frame and blocks. Furthermore, these loads may cause structural failure in the connection, such as relative radial slipping between the blocks and docking frame, resulting in gaps between docking frames. To prevent structural damage, the docking frame is usually made from high-strength forged aluminum rings, while the blocks undergo precision processing after aluminum forging. Additionally, it is essential to arrange the material's fibers along the circumferential direction to enhance its performance. To prevent structural failure, increasing the pre-tensioning force of the clamp band is commonly employed.

Before the rocket launch, the explosive bolts are gradually tightened to apply an appropriate circumferential pre-tensioning force to the clamp band. This action secures the blocks radially to fasten the interstage connection, completing the connection of the stages. During the loading of the pre-tensioning force, the strain on the surface of the clamp band is monitored, and the pre-tensioning force of the clamp band is calculated based

on the measured strain values to determine whether the pre-tensioning force is uniform across the clamp band. Additionally, as the clamp band is fastened to the satellite frame, the uniformity of the applied pre-tensioning force can also be judged based on the stress condition of the satellite frame by monitoring the strain information on the satellite frame.

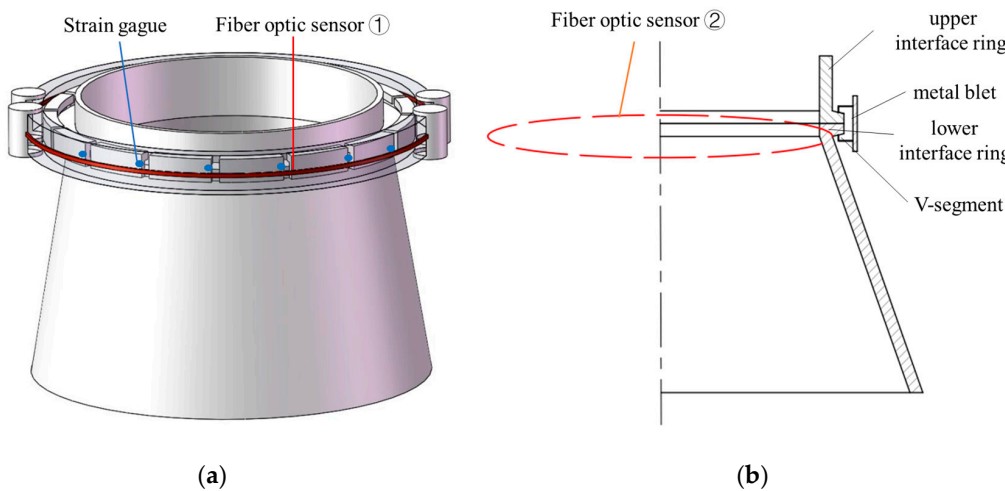

(**a**)  (**b**)

**Figure 1.** Configuration of clamp band joint. (**a**) Primary view; (**b**) section view.

### 2.2. The Foundation of Fiber Optic Monitoring

When light travels through optical fibers, external factors like temperature and strain can induce alterations in light propagation. Distributed optical fibers detect variations in measured physical quantities by analyzing changes in light signals. Rayleigh scattering, an inherent and stable scattering mode within each optical fiber, primarily exploits density non-uniformities in the glass molecules. These non-uniformities lead to varying refractive indices for light, thereby altering the wavelengths of light in Rayleigh scattering. If deformation occurs at a specific position in the optical fiber, the reflected light wavelength at that point will shift. By comparing the reflected light before and after deformation, it is possible to pinpoint which part has undergone changes [23–25], as shown in Figure 2.

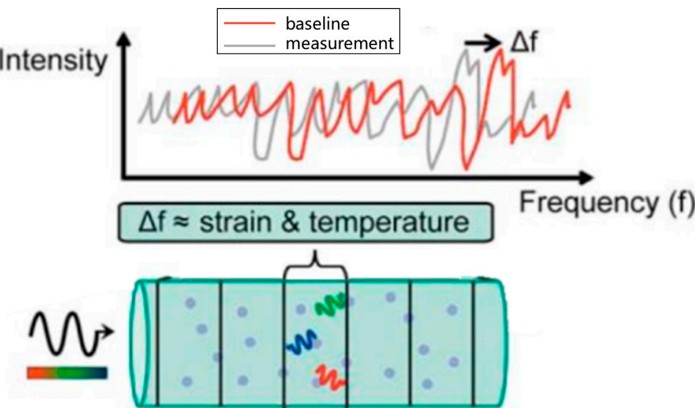

**Figure 2.** Measurement principle of a distributed optical fiber.

Leveraging wavelength scanning interferometry technology enables precise and practical distributed measurements of temperature and strain along optical fibers spanning from several tens to over a hundred meters in actual length, achieving millimeter-level spatial resolution. Strain and temperature resolutions can be finely tuned to 1 με and 0.1 °C, respectively. These accuracies are attributed to the correlation between temperature and strain variations and alterations in the scattered light spectrum within the optical fiber [26].

The spectral changes $\Delta\lambda$ that are responsive to strain $\varepsilon$ or temperature $T$ can be expressed as follows:

$$\frac{\Delta\lambda}{\lambda} = K_T\Delta T + K_\varepsilon\varepsilon \tag{1}$$

where $\lambda$ represents the average wavelength of the light wave, and $K_T$ and $K_\varepsilon$ are the standard constants for temperature and strain, respectively. Typically, the values of $K_T$ and $K_\varepsilon$ exhibit a 10% variation compared to the standard communication optical fibers. The default values for these constants are set at values common for most germanosilicate core fibers: $K_T = 6.45 \times 10^{-6}\,°\mathrm{C}^{-1}$ and $K_\varepsilon = 0.780$.

### 2.3. Strain Monitoring Method for Uniform Preload

To enable real-time monitoring and analysis of the uniformity of the pre-tensioning force applied to the clamp band, it is necessary to deploy distributed optical fiber sensors on both the clamp band and the satellite frame. The band is adjusted by tapping and loosening the clamping block. This allows for the real-time monitoring of strain information during the application of the pre-tensioning force. Prior to monitoring, it is essential to mark the positional information of each block. The method proposed in this paper for monitoring and determining the uniform application of pre-tensioning force in the interstage structure involves a coarse adjustment when the pre-tensioning force reaches the expected load. Initially, based on the strain data of the clamp band, extract the strain data of each block and take their average value as the reference value. Identify the block with the greatest deviation from the average value, adjust the tightness of that block (using a torque wrench for adjustments), re-measure the strain of the clamp band after the adjustment, and obtain a new average value. Repeat this process until the strain values at each position deviate from the average value by less than 10%, which concludes the coarse adjustment. Subsequently, using a similar method, fine-tune the clamp band based on strain data from the satellite frame. Based on previous experience, the strain of the satellite frame is approximately one-tenth that of the clamp band. Hence, during the fine-tuning process, the adjustment of block tightness or tapping should be one-tenth of that in the coarse adjustment phase. For instance, if in the coarse adjustment phase, the torque wrench is rotated by one full turn, in the fine-tuning phase, the torque wrench should be adjusted by one-tenth of a turn. Additionally, the deviation of the average strain value of the satellite frame can be twice that of the deviation of the clamp band, such as 15%. The workflow of this method is depicted in Figure 3.

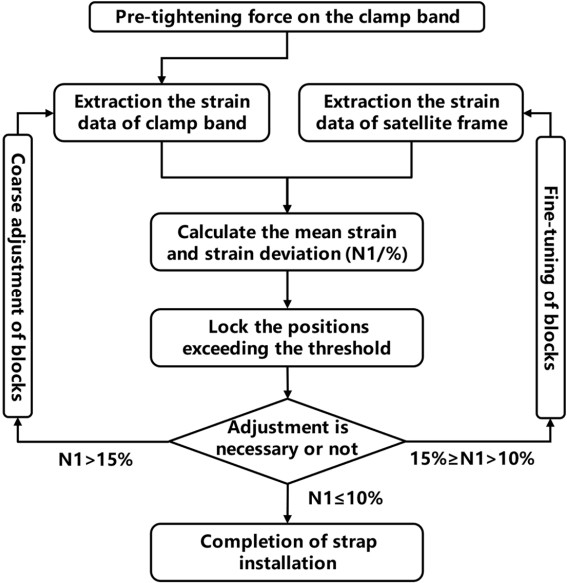

**Figure 3.** Flowchart of uniform preload adjustment based on strain monitoring data.

### 3. Finite Element Model

In this section, ABAQUS 2020 software is used to conduct a finite element simulation of the satellite clamp band structure, and the strain characteristics of the structure are analyzed through a simulation analysis. Since the satellite clamp band structure is a periodic cycle structure, the structure can be simplified and 1/10 of the structure was selected for the simulation calculations. As shown in Figure 4, the model includes the satellite frame, clamp band, and two card blocks. The boundary of the satellite frame is constrained by periodic boundary conditions. When the satellite and rocket locking device is a connected structure of mechanism, the load is transferred through contact and friction between the components. A contact analysis was adopted to study the relationships between the satellite frame, the clamp block, and the tape, and the interface friction coefficient μ = 0.3. The influence of the satellite's mass was not taken into account in the modeling, where the model consisted of 34,886 units. The material properties of each part of the model are shown in Table 1. The circumferential force along the belt was applied to the annular section to simulate the preload force during the installation of the structure.

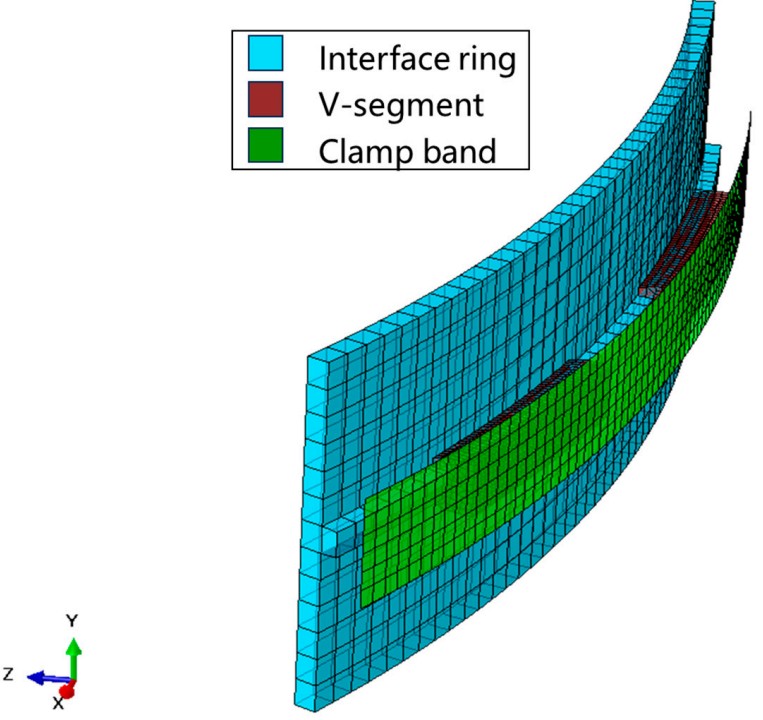

**Figure 4.** Finite element model.

**Table 1.** Material properties of the model.

| Part(s) | Material | *E*/GPa | μ |
|---|---|---|---|
| Interface ring and V-segment | Aluminum alloy | 70 | 0.3 |
| Clamp band | High-tensile steel | 160 | 0.3 |

The stress and strain cloud diagram of the finite calculation results are shown in Figure 5, where the strain value is the circumferential strain in the polar coordinate system with the structure center as the origin. The strain curves obtained by extracting the circumferential strain at the center line of the structure between the inner side of the satellite frame and the outer side of the tape are shown in Figure 6.

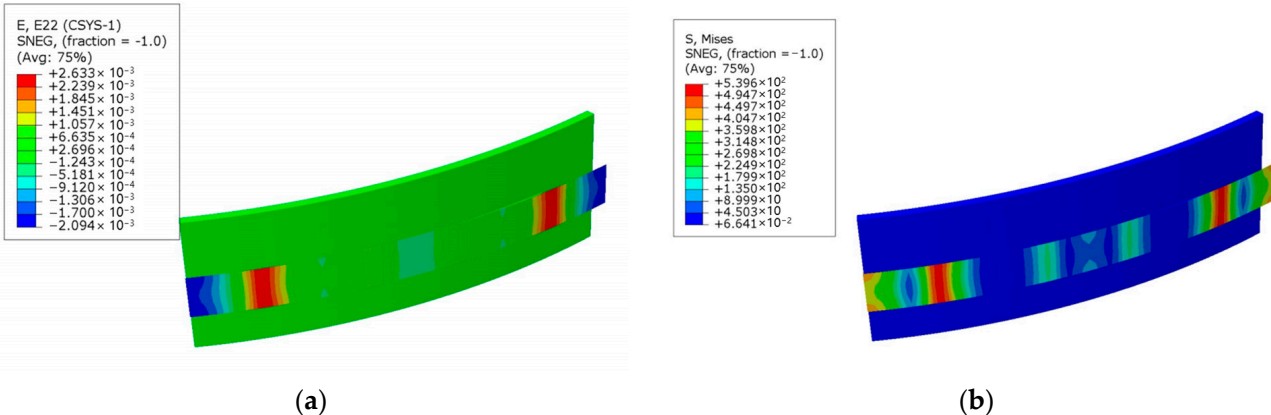

(**a**)                                                                 (**b**)

**Figure 5.** Simulation results of the clamp band. (**a**) Stress cloud map; (**b**) strain cloud map.

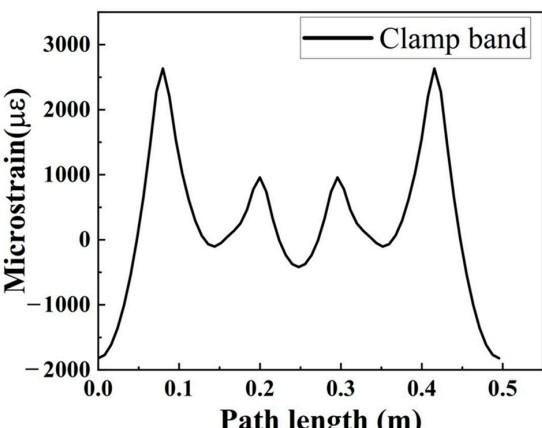

**Figure 6.** The radial strain distribution at the midline of the clamp band.

As shown in Figure 6, when the tape is strained, the strain between two clips is the smallest, the strain at the edge of the clip is the largest, and the strain at the middle of the clip is smaller than that at the edge.

## 4. Experiment Test

### 4.1. Test Setup

The fiber optic sensing network of the satellite–rocket connection device includes two distributed optical fiber sensors: one is installed on the outer side of the clamp band, and the other is placed at the edge of the docking frame, as shown in Figure 1. In order to compare the monitoring data between the optical fibers and strain gauges, corresponding strain gauges were placed at the respective positions of each block on the outer side of the clamp band. According to the simulation results in the last paragraph of Section 3, the strain gauge was arranged at the edge of each clamp block, that is, the position where the strain is large. During the installation of the clamp band, the gradual loading of pre-tensioning force was performed, as illustrated in Table 2. The strain data from both the optical fibers and strain gauges were collected using the LUNA(USA)-ODiSI A50 equipment and Donghua (China) strain monitoring instrument. The experimental setup employed a Rayleigh scattering-based distributed optical fiber sensor developed by Luna Innovations. This sensor is equipped with a polyimide outer coating and has an external diameter of 155 μm, facilitating continuous strain measurements at the millimeter scale. The optical fiber attached to the surface was initially secured at the endpoint using 401 quick-drying adhesives, with the entire path covered and protected by Araldite® (China) 2011 epoxy adhesive. The adhesive used for the strain gauges was identical to that of the optical fiber.

**Table 2.** Preload setting.

| Loading Level | 1 | 2 | 3 | 4 | 5 | 6 | 7 | 8 | 9 | 10 | 11 |
|---|---|---|---|---|---|---|---|---|---|---|---|
| Preload/kN | 0 | 4.65 | 8.02 | 10.59 | 11.96 | 13.38 | 15.24 | 16.63 | 22.46 | 23.65 | 25.57 |

For all experiments and measurements described below, the optical backscatter reflectometer ODiSI A50 by Luna Innovations was applied in Figure 7. Table 3 provides a summary of the crucial measuring parameters for the data analysis.

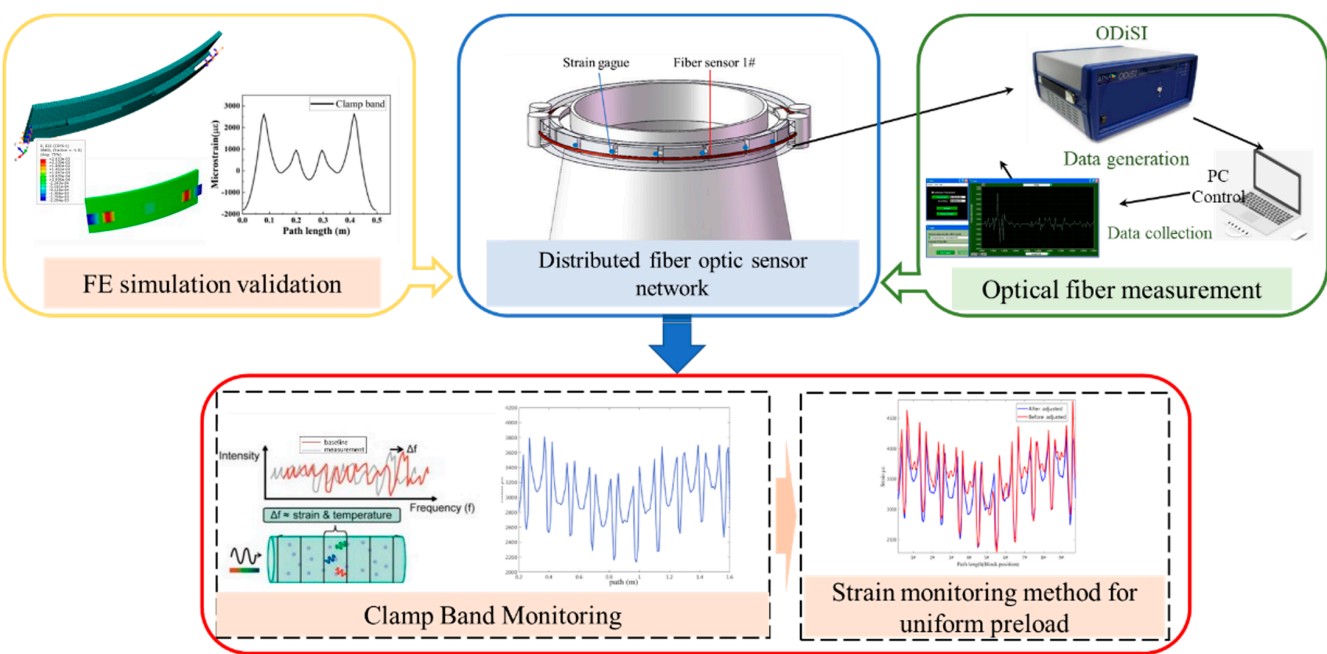

**Figure 7.** Strain monitoring and installation adjustment of satellite–rocket connection device based on a distributed optical fiber.

**Table 3.** Acquisition system parameter settings.

| Parameter | Value |
|---|---|
| Distance range (standard mode) | up to 50 m |
| Strain range | $\pm 5000$ $\mu\varepsilon$ |
| Spatial resolution (gauge spacing) | 5 mm |
| Gauge length | 10 mm |
| Strain measurement resolution | $\pm 1$ $\mu\varepsilon$ |

In the actual strain measurements, we can only obtain the average deformation of the structure within a certain distance. The primary factors affecting strain demodulation results include the signal quality of the collected Rayleigh scattering spectrum and the algorithmic strategy for demodulating the Rayleigh scattering spectrum. The signal quality of the collected Rayleigh scattering spectrum is mainly determined by the hardware components of the OFDR system. Generally, the higher the signal-to-noise ratio of the collected Rayleigh scattering spectrum, the better the signal quality, leading to higher reliability and accuracy in demodulation. The core of the demodulation algorithm is based on the cross-correlation function to analyze the offset of the Rayleigh scattering spectrum. Therefore, factors such as the selection of the shape function, the subset window length, and the criteria for the cross-correlation function in the demodulation algorithm all influence the precision of the cross-correlation calculation. For the reasons mentioned above, during the debugging phase of the pre-tensioning force loading test, sampling was conducted at

gauge lengths of 2 mm, 5 mm, and 10 mm. From Figure 8a, it is evident that a 10 mm gauge length led to missing critical data. Meanwhile, in Figure 8b, the data demodulation at a 2 mm gauge length showed additional noise and demodulation errors. Consequently, in this experiment, a gauge length of 5 mm was selected.

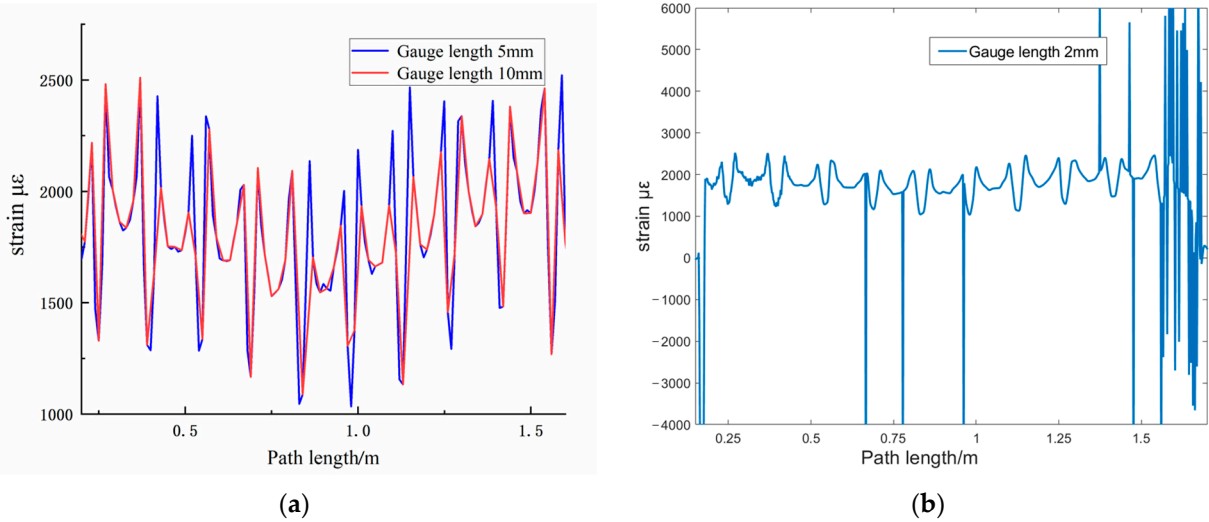

(**a**)　　　　　　　　　　　　　　　(**b**)

**Figure 8.** Strain curves at different gauge length. (**a**) Gauge lengths of 5 and 10 mm; (**b**) gauge length of 2 mm.

### 4.2. Effect of Preload

The left and right halves of the clamp band exhibited symmetric strain distributions. This analysis was based on the data from the left half of the clamp band. There are nine blocks on the left half of the clamp band, with each block having a strain gauge installed at its position. Here, the strain information from the edge of the clamp band (#1) and the middle (#4) is listed. As shown in Figure 9, the strain data from both the fiber optic and strain gauges exhibited similar trends, showing linear strain growth with an increase in pre-tensioning force. The average deviations between the two strain gauges and the fiber optic data were 2.15% and 2.85%, respectively. The largest deviation occurred at the maximum pre-tensioning force of 25 kN, reaching 4.23%. The consistency between the measurement results obtained from the fiber optic and strain gauges is satisfactory.

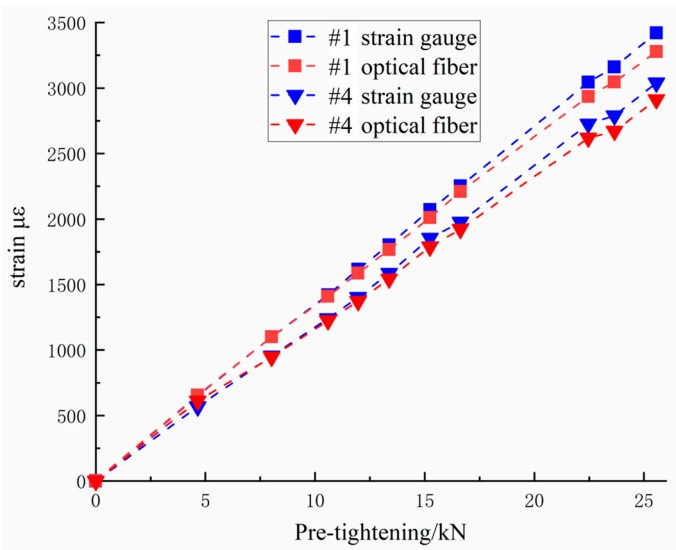

**Figure 9.** Comparison of strain data from optical fiber and strain gauges.

The strain distribution of the left half of the clamp band under a gradually increasing pre-tensioning force is shown in Figure 10: the clamp band was under tension, and there was a similar distribution trend at each block position. Figure 10b compares the measured pre-tightening force values of the clamp band with the finite element results. The finite element model was simplified, but the periodic strain trends of each block were still evident. The strain between adjacent blocks was relatively small, with a larger strain at the edges of each block and smaller strain in the middle positions. With the incremental increase in pre-tensioning force, the strains at all locations showed a growing trend. The maximum strain occurred under the maximum pre-tensioning force condition, reaching approximately 4500 microstrains at both ends of the blocks. At this point, there was a difference of about 1500 microstrains between the maximum and minimum strains.

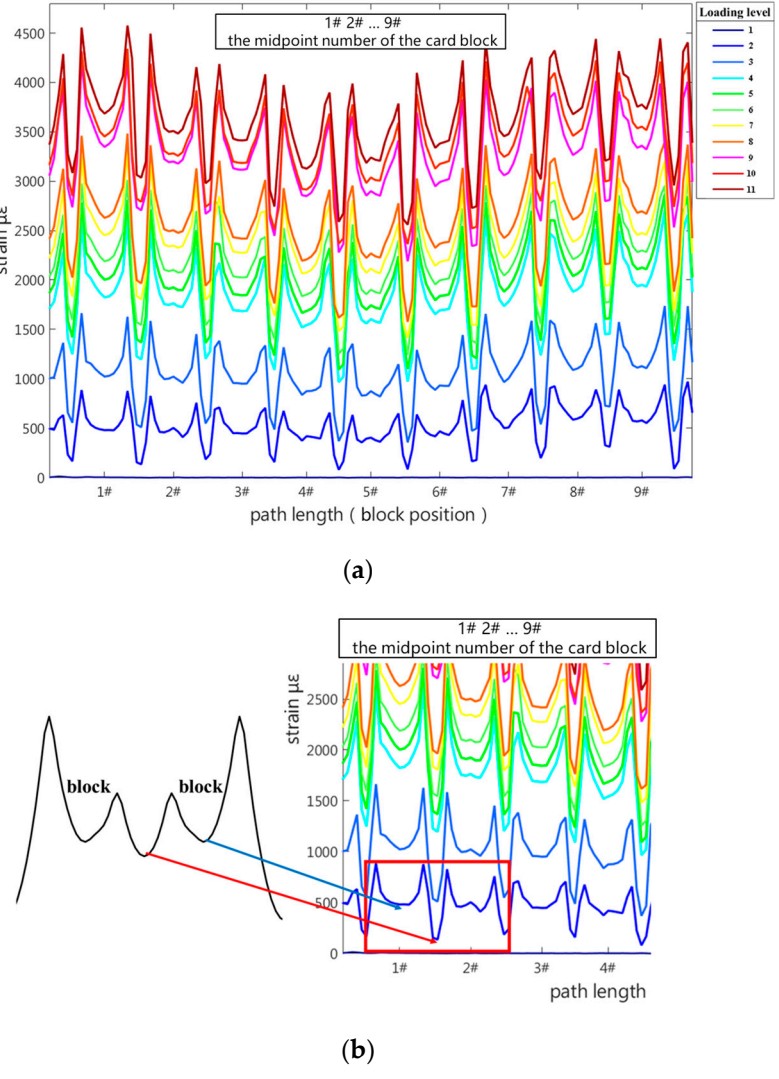

(**a**)

(**b**)

**Figure 10.** Variation in circumferential strain of the clamp band under preload force. (**a**) Strain data under preload; (**b**) comparison with finite element results.

The left and right halves of the satellite frame exhibited a symmetric strain distribution. Here, the analysis was performed on the data from the left half of the satellite frame, which corresponds to the clamp band. The strain distribution of the left half of the satellite frame under a gradually increasing pre-tensioning force is depicted in Figure 11: the satellite frame was under compression, and there was a similar distribution trend at each block position. The compressive strain between adjacent blocks was relatively small, with a larger compressive strain at the middle positions of each block. With the incremental

increase in the pre-tensioning force, the compressive strains at all locations exhibited a growing trend. The maximum compressive strain occurred under the condition of the maximum pre-tensioning force, reaching close to 500 microstrains, with a difference of about 300 microstrains between the maximum and minimum strains.

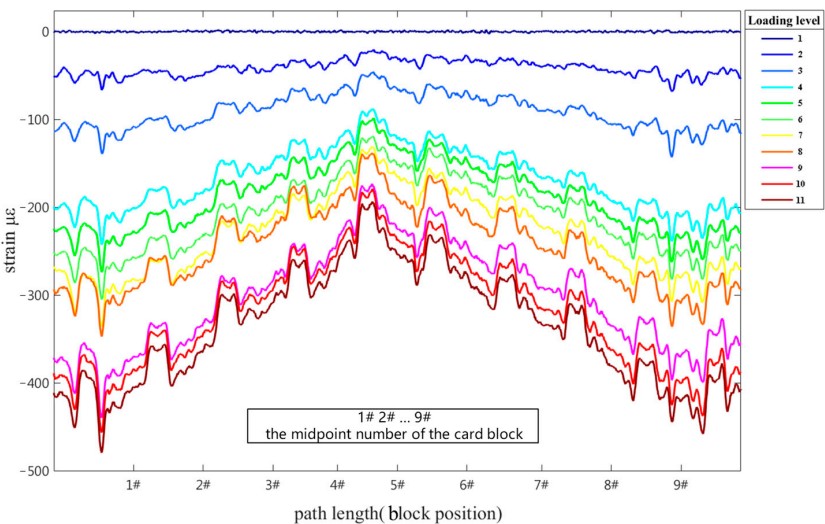

**Figure 11.** Variation in circumferential strain of the satellite frame under preload.

### 4.3. Adjustment Result

When the pre-tensioning force reached 22.5 KN in the experiment, adjustments were made for the uniform distribution of the pre-tensioning force, as shown in Figure 12. The strain data for each block on the clamp band were extracted at this point. The average strain for each block was calculated, and block #2 was identified as having the highest deviation of 14.1% from the overall average. Following the process outlined in Figure 3, step-by-step coarse and fine adjustments were carried out. The strain data for each block before and after adjustment are presented in Table 4. The standard deviation of the strain data for each block on the clamp band decreased from 219 to 174. After adjustment, the maximum deviation of the block position from the mean was 8.11%, meeting the adjustment standard for this method. Figures 13b and 14, respectively, provide the strain comparison before and after the adjustment of the clamp band and satellite frame under the same pre-tightening force. The uniformity of the strain data improved compared to before the adjustments.

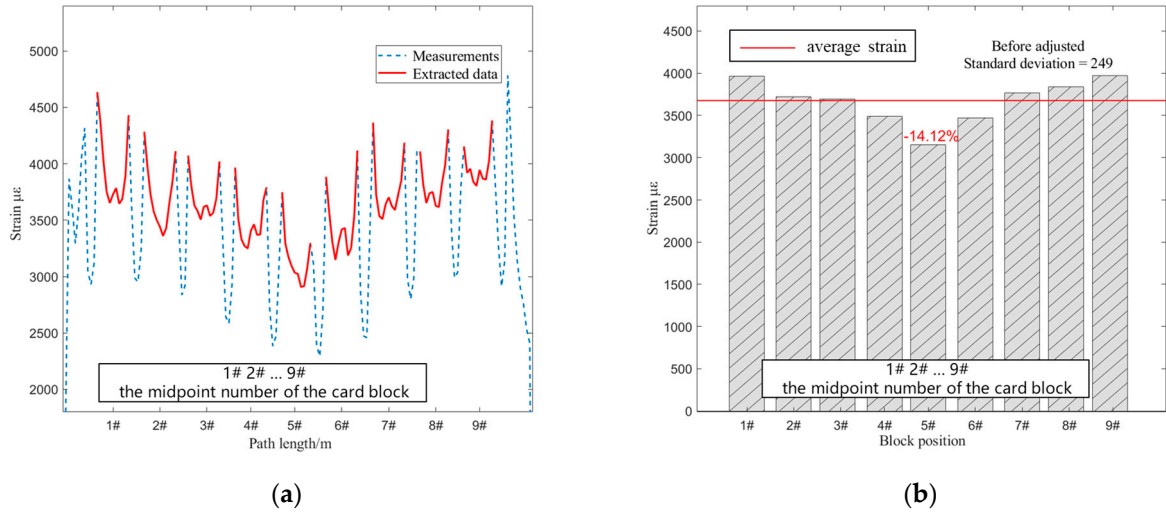

**Figure 12.** Data processing method for uniformly distributed strain. (**a**) Extraction of clamp band strain data; (**b**) mean strain data for each block position.

**Table 4.** Strain data before and after adjustment of the clamp band (unit: με).

| Number of Block | | #1 | #2 | #3 | #4 | #5 | #6 | #7 | #8 | #9 | Mean Value | Standard Deviation |
|---|---|---|---|---|---|---|---|---|---|---|---|---|
| Before adjustment | Mean strain | 3962 | 3718 | 3696 | 3491 | 3156 | 3468 | 3768 | 3842 | 3947 | 3675 | 249 |
| | deviation (%) | 7.82 | 1.16 | 0.57 | −5.01 | −14.1 | −5.64 | 2.53 | 4.55 | 8.14 | | |
| After adjustment | Mean strain | 3761 | 3520 | 3436 | 3241 | 3242 | 3433 | 3587 | 3752 | 3773 | 3527 | 197 |
| | deviation (%) | 6.62 | −0.22 | −2.6 | −8.11 | −8.07 | −2.67 | 1.69 | 6.38 | 6.96 | | |

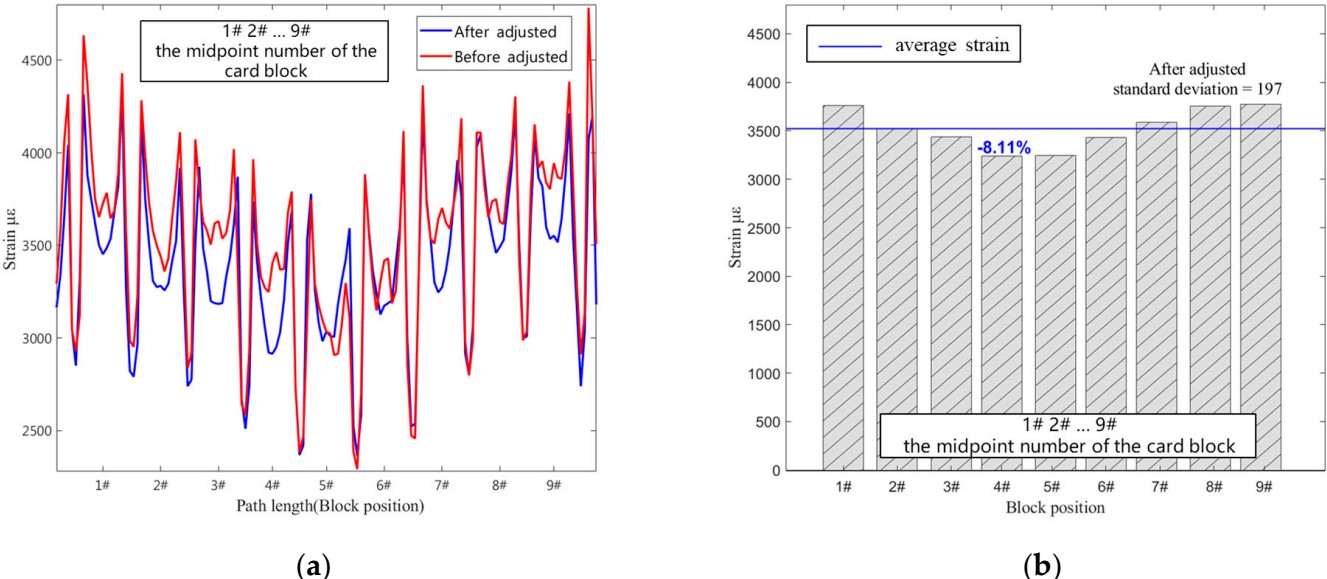

(**a**)  (**b**)

**Figure 13.** Comparison of strain data before and after adjustment. (**a**) Mean values at each block position; (**b**) comparison of before and after adjustment.

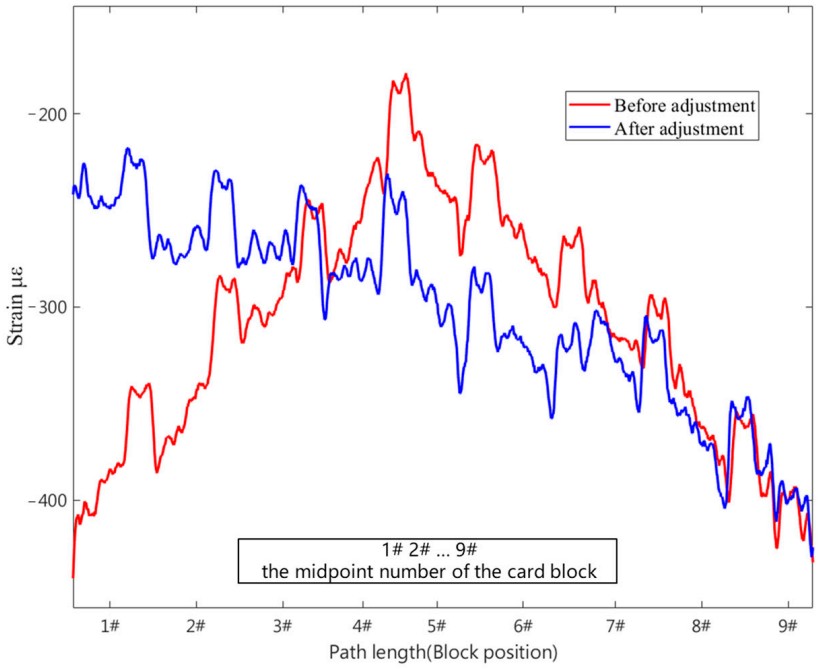

**Figure 14.** Comparison of strain data of the interface ring before and after adjustment.

## 5. Conclusions

This paper proposed a method assisted by strain data for the installation of a satellite–rocket connection device in cases of uneven application of the pre-tensioning force during installation.

By using finite element software, the strain distribution of the tape and satellite frame under a preload was simulated. The strain of the structure changed periodically: the tape was strained, the strain was the largest at the corresponding edge of the block, and the strain between the block was the smallest. When the satellite frame was compressed, the strain at the edge of the corresponding contact block reached the maximum, and the strain between the block was at the minimum.

Distributed optical fiber sensors were used to monitor in real-time the strain on the clamp band of the satellite–rocket connection device and satellite frame during the incremental application of a pre-tensioning force in the interstage connection device. The fiber optic data on the clamp band and the strain gauge data exhibited excellent consistency.

The strain monitoring data for both the clamp band and satellite frame demonstrated periodic variations in strain caused by discontinuities in stiffness due to the blocks: a significant strain appeared between blocks, while a relatively smaller strain occurred at the block positions. Based on the materials of the clamp band and satellite frame and the maximum strain, it was concluded that both were within the allowable strength range. This is in agreement with the finite element simulation results.

In addressing the issue of uneven pre-tensioning forces, applying this method for strain monitoring and adjusting the blocks allowed for a relatively uniform pre-tensioning force. The standard deviation of the strains for each block was reduced after adjustments. The experiment validated the feasibility of using strain monitoring to assist in the installation of the interstage connection device, providing guidance for the even application of the pre-tensioning force during installation.

**Author Contributions:** Conceptualization, Z.Y., L.Y. and Z.W.; Methodology, X.Q., S.Z., F.M., J.L., H.X. and L.Y.; Validation, H.X.; Investigation, X.Q., S.Z., F.M., J.L. and H.L.; Resources, Z.Y.; Data curation, X.Q.; Writing—original draft, X.Q. and S.Z.; Writing—review and editing, Z.Y. and L.Y.; Supervision, Z.W.; Funding acquisition, L.Y. and Z.W. All authors have read and agreed to the published version of the manuscript.

**Funding:** This work was supported by the National Key Research and Development Program of China (Grant Nos. 2022YFB3402500 and 2018YFA0702800), Young Elite Scientists Sponsorship Program by CAST (2022QNRC001), and Dalian Youth Science and Technology Star: 2023RQ004.

**Data Availability Statement:** The data that support the findings of this study are available on request from the corresponding author. The data are not publicly available due to privacy.

**Conflicts of Interest:** The authors declare no conflict of interest.

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
