# Peer review of "Strain Monitoring and Installation Adjustment of Satellite–Rocket Connection Device Based on Distributed Optical Fibers"

_photonics, doi:10.3390/photonics11040335_

Round 1

Reviewer 1 Report

Comments and Suggestions for Authors

The monitoring method proposed in this paper is innovative. The expression is clear and the experimental design and data analysis are appropriate. I recommend the publication of this article, following the correction of the issues and formatting errors listed below.

1. The introduction lacks a discussion on the research progress of other sensors monitoring this structure.

2. In Section 2.1, the introduction of the structure is excessively detailed.

3. In Section 2.2, OFDR, as a common method monitoring, can be briefly introduced in terms of its working principle.

4. In Line 207, 208 and 210, the word “envelope” should change to “clamp band”.

5. In Fig.4, Is it possible to match the angle of the model display with that of Fig.5

6. The format of Fig.8, 9, 12 and 14 should be unified.

7. Is it necessary to enumerate the strain information at each position in Table 4? The content is duplicated in Fig12 and 13.

Reviewer 2 Report

Comments and Suggestions for Authors

This manuscript discusses the employment of distributed optical fiber sensor for strain monitoring and installation adjustment of satellite-rocket connection device under preload. Overall, this work is interesting and has practical applicability. I recommend the following improvements before it can be accepted for publication.

1)     The full name of the abbreviations should be given, for instance, INTA, OPTOS, LIDS.

2)     In Line 103, before saying “to address issues with traditional strain gauge monitoring methods …”, the research status of the traditional strain gauge monitoring methods should be summarized.

3)     In Fig.1, the “Fiber sensor” should change to “Fiber optic sensor”.

4)     In Fig.4, it is suggested to use different colors for different parts of the finite element model, for better clarity.

5)     The titles of Figs. 7, 12 and 13 need to modify to better express the meaning of the figures.

6)     In Table 3, there is no need to add a column for the units. And the unit of strain measurement resolution is mm?

7)     Line 313, the sentence repeats with above.

8)     In Table 4, there is a lack of unit.

Comments on the Quality of English Language

The English should be improved.

Reviewer 3 Report

Comments and Suggestions for Authors

The paper proposes a monitoring method aimed at the uniform distribution of pretension. The manuscript is well organized. I recommend this article for publication, provided that the authors successfully address the queries below and undertake a revision of the manuscript to correct a handful of typographical errors.

1.     For this structure, are there any other monitoring methods that can be supplemented in the introduction?

2.     In Fig.1 the labeling of the sensors as "1#" and "2#" is the same as the marking method for the blocks, which can easily lead to confusion.

3.     In Fig.3, is the setting of the method's thresholds (10%, 15%) based on structural requirements or determined by testing experience?

4.     The model in Fig.4 is not clearly displayed.

5.     In section 4.1the detailed information of sensor (e.g. sensor type) and its installation (e.g. bonding material) should be included.

6.     In Fig.8, the specifications (color, size, etc.) of the two charts are not unified.

7.     In Section 4.2, is it necessary to display the results under the 0kN load?

8.     In Table 4, the data format for "Deviation" is incorrect and not aligned.

Reviewer 4 Report

Comments and Suggestions for Authors

This article discusses the utilization of optical frequency domain reflectometry (OFDR) and strain gauges for measuring strain distribution, thereby guiding the adjustment of the satellite-rocket connection device to ensure uniform preload distribution. The overall structure of the article is good. However, I have several concerns and queries that necessitate further clarification from the authors, leading me to recommend a major revision.

1. The fiber optic technique used in this study, namely OFDR, should be detailed in both the Abstract and the Introduction. Furthermore, Section 2.2 requires revision to describe the OFDR principle more accurately and to prevent duplication of content in another article by the authors titled “Various static loading condition monitoring of carbon fiber composite cylinder with integrated optical fiber sensors” in Optical Fiber Technology (vol. 83, art. no. 103685, 2024).

2. Following the first question, I would like the authors to explain the novelty of this article as compared to their published paper in Optical Fiber Technology. From my perspective, the same fiber optic sensing technique, strain gauges, finite element analysis, and comparison methods are used in both papers.

3. Concerning the markings in Figures 1 and 9, the hash symbol (#) should precede the numbers for clarity.

4. A friendly reminder regarding potential copyright issues concerning Figures 2 and 7, which appear in the Luna ODiSI user's manual and a 2016 conference paper titled "Parameter identification based on quasi-continuous strain data captured by high resolution fiber optic sensing." The copyright issue should be addressed carefully.

5. It is recommended to cite a reference for Equation (1) and to provide the values of the constants used within.

6. In Line 243, there is an error in the section index which needs correction.

7. The specific sensing fiber used in this study should be provided for repeatable experiments.

8. Clarification is required regarding the statement in Line 259, where the measurement range is described in terms of the grating length in fiber optic gratings. However, the study does not utilize gratings.

9. In line 261, OFDR does not yield more accurate results with a smaller gauge length.

10. In line 267, the full name of OFDR should be provided before the abbreviation is used.

11. I recommend replacing the term “measurement point interval” with “gauge length” to avoid confusion, as these concepts are not identical in OFDR.

12. In the legend of Figure 14, “adjusted” should be changed to “adjustment”.

13. Various fiber optic sensing techniques such as fiber Bragg grating and Fabry-Pérot cavity are available for strain measurements in a more economical manner. What motivated the selection of OFDR in this study for strain monitoring? Incorporating insights from additional sources would be beneficial for the readers to better understand the choice (suggested article: 10.1109/JSEN.2023.3343604, 10.1109/JSEN.2022.3197730).

Comments on the Quality of English Language

The English usage is generally okay, but it would be beneficial to conduct a thorough review to avoid grammatical errors.

Round 2

Reviewer 4 Report

Comments and Suggestions for Authors

I would like to say thanks to the authors for their efforts in revising the manuscript. Most of the inquiries were addressed properly. The article can be accepted for publication, but I suggest the authors either redraw Figures 2 and 7 or cite appropriate references for these figures to avoid any copyright conflicts. The relevant source links are provided below.

https://www.lambdaphoto.co.uk/pdfs/LUNA/Article%20-%20Distributed%20fibre-optic%20temperature%20and%20strain%20measurement%20withextremely%20high%20spatial%20resolution.pdf

https://magazine.polytec.com/eu/distributed-fiber-optic-temperature-and-strain-measurement

Please consider and use the comments as my support to improve your paper. Furthermore, I appreciate the authors’ ongoing contributions to the field of engineering applications in fiber optic sensing. I do look forward to your future publications on the evaluations of optical fiber sensor installations using different adhesives and solvents.

Comments on the Quality of English Language

Generally good.